# Sequence of antihypertensive medications used in preterm infants with hypertension: A cross-sectional study

Alejandro D. Perez[1¤a], Mary-Carty Pittman[1¤b], Kaniz Afroz Tanni[2], Keia R. Sanderson[3], Jieun Park[4], Daniel I. Feig[5], Matthew M. Laughon[6], Matthew Shane Loop[1,2*]

1 Division of Pharmacotherapy and Experimental Therapeutics, UNC Eshelman School of Pharmacy, University of North Carolina at Chapel Hill, Chapel Hill, North Carolina, United States of America, 2 Department of Health Outcomes Research and Policy, Harrison College of Pharmacy, Auburn University, Auburn, Alabama, United States of America, 3 Division of Nephrology and Hypertension, Department of Medicine, University of North Carolina School of Medicine, University of North Carolina at Chapel Hill, Chapel Hill, North Carolina, United States of America, 4 Harrison College of Pharmacy, Auburn University, Auburn, Alabama, United States of America, 5 Department of Pediatrics, Marnix E. Heersink School of Medicine, University of Alabama at Birmingham, Birmingham, Alabama, United States of America, 6 Division of Neonatal-Perinatal Medicine, Department of Pediatrics, University of North Carolina School of Medicine, University of North Carolina at Chapel Hill, Chapel Hill, North Carolina, United States of America

¤a Current address: Department of Pharmacy, University of North Carolina Medical Center, Chapel Hill, North Carolina, United States of America
¤b Current address: Department of Pharmacy, WakeMed, Raleigh, North Carolina, United States of America
* matthew_loop@auburn.edu

## Abstract

### Background

Hypertension in preterm infants can result in life-threatening outcomes. However, there is limited evidence to guide the pharmacologic management of hypertension in preterm infants. Without population-level studies of the pharmacologic strategies that clinicians currently employ for hypertension in preterm infants, studies investigating the benefits and risks of these strategies cannot be performed.

### Methods

A retrospective, cross-sectional study was conducted at a single academic medical centerto determine the most prevalent antihypertensive medication used for first-line, second-line, and adjunctive pharmacologic management among preterm infants with hypertension. The study sample included patients with a postnatal age less than 1 year at hospital discharge, gestational age at birth less than 37 weeks, and treated with an antihypertensive medication between July 2010 and December 2022. The prevalence of each antihypertensive medication used for each stage of pharmacologic management was estimated, and prevalences were compared using prevalence ratios. Bayesian multinomial regression was used to estimate prevalence ratios and 95% credible intervals (CIs).

**Data availability statement:** This study was determined to be Not Human Subjects Research by Auburn University and the University of Alabama at Birmingham. The data from these electronic medical records was delivered from the University of Alabama at Birmingham to Auburn University on January 4th, 2023. Study investigators never had access to information that could identify participants, either during or after data collection. The study authors are prevented from sharing the data publicly due to restrictions placed upon use of the data by the data owners. To initiate a request for the dataset, please contact ResearchData@uabmc.edu and reference request "i2b2.lds.inpatient infants receiving antihypertensives w/dx codes <37 pt set". The corresponding author (matthew_loop@auburn.edu) will do everything he can to facilitate obtainment of the data for reasonable requests.

**Funding:** The research reported in this manuscript was supported by the National Center for Advancing Translational Sciences of the National Institutes of Health [grant number UL1TR003096 to the University of Alabama at Birmingham]. The content is solely the responsibility of the authors and does not necessarily represent the official views of the National Institutes of Health. The funder had no role in study design, collection, analysis, interpretation of data, writing the report, or the decision to submit the article for publication.

**Competing interests:** The authors have declared that no competing interests exist.

## Results

Out of 751 possibly eligible patient encounters obtained, 134 encounters were identified across 120 patients that met the inclusion criteria. Second-line and adjunctive pharmacologic management were used in 6 and 12 encounters, respectively. Propranolol had the highest prevalence for each stage of pharmacologic management: 61% for first-line (95% CI: 53-69%); 40% for second-line (95% CI: 14-70%); and 47% for adjunctive pharmacologic management (95% CI: 24-72%). As a first-line pharmacologic management, propranolol was 4.8 times (95% CI: 2.9-7.7) as prevalent as the second most prevalent medication (captopril).

## Conclusion

In a large academic hospital, propranolol was the most commonly used medication for first-line antihypertensive pharmacologic management among preterm infants. The most prevalent medication used for second-line and adjunctive pharmacologic management was inconclusive.

## Introduction

Hypertension occurs in approximately 2% of Neonatal Intensive Care Unit (NICU) admissions [1]. It is defined as systolic blood pressure above the 95th percentile (for patient age/weight) on three occasions [1,2]. Preterm infants (gestational age at birth less than 37 weeks) have an increased risk of developing hypertension because they are predisposed to having a low birth weight and experiencing sudden exposure to the extrauterine environment at a time when organogenesis is incomplete [3,4]. Among preterm infants, bronchopulmonary dysplasia, and iatrogenic factors are other risk factors associated with the development of hypertension [5]. The complications of untreated, severe hypertension in infants include end-organ damage, such as vascular injury, encephalopathy, hypertensive retinopathy, cardiomyopathy, and mortality [6]. Therefore, proper management of hypertension in preterm infants is imperative to optimize clinical outcomes.

Unfortunately, the evidence available to clinicians to guide the pharmacologic management of preterm infants with hypertension is minimal. There are no guidelines or formal statements from professional organizations on this topic. The primary literature does not include quantitative safety or efficacy data for antihypertensive treatments in preterm infants. Several review articles have described the general diagnostic approach to identifying neonatal hypertension and summarized the pharmacokinetic properties, side effect profiles, and dosing strategies associated with commonly prescribed antihypertensives to treat the disease [7–10]. A multi-center study by Ravisankar et al. (2017) investigated antihypertensive medication exposure in preterm infants from 1997 to 2013 and found that at least 12 different medications were used to treat hypertension in preterm infants [11]. Overall exposure to these antihypertensive medications also varied widely across sites [11]. However, the sequence in which antihypertensive medications are used is still poorly understood. Therefore, the objective of this study was to determine the sequence of antihypertensive medications used to treat hypertension in preterm infants at a large medical center.

## Materials and methods

### Study population

The target population for this retrospective, cross-sectional, observational study was preterm infants who were treated for hypertension. To sample from this population, we identified

preterm infants admitted to the University of Alabama at Birmingham (UAB) Hospital. The UAB Data Warehouse and the i2b2 database (an informatics tool that organizes electronic medical record data) [12] were used to identify all patients admitted to the UAB medical center between July 29, 2010, and December 31, 2022 who met the following inclusion criteria: 1) gestational age at delivery < 37 weeks; 2) postnatal age < 1 year upon hospital discharge; and 3) received at least one of the following medications during the inpatient stay: hydralazine, sodium nitroprusside, clonidine, captopril, enalapril, propranolol, labetalol, esmolol, atenolol, metoprolol, amlodipine, isradipine, nifedipine, nicardipine. We excluded hospital stays with a discharge diagnosis of hemangioma in any position because hemangioma is an alternative indication for propranolol.

## Outcomes

This study had three outcomes. First-line pharmacologic management with an antihypertensive medication was defined as the first administration of any antihypertensive medication. Second-line pharmacologic management was defined as the administration of a second antihypertensive medication after the initial antihypertensive medication had been discontinued within the same hospitalization. Adjunctive pharmacologic management was defined as the administration of two distinct antihypertensive medications within 24 hours of one another on at least two occasions.

## Covariates

Patient characteristics included sex, gestational age at birth (weeks), age at discharge (days), race (Asian, Black or African American, Hispanic or Latino, Other, and White), ethnicity (Hispanic/Latino or Non-Hispanic/Latino), birthweight (grams), and percent of systolic blood pressure measurements above the 95th percentile in the 24 hours before initial antihypertensive treatment. We used percentiles for infants at two weeks postnatal age based on Table 2 of Dionne, Abitbol, and Flynn (2012) [9]. Once a patient's PMA passed 44 weeks, we assigned blood pressure percentiles according to the Second Task Force on Blood Pressure Control in Children - 1987 report [13].

## Statistical analysis

Medians, first quartiles, and third quartiles were calculated for continuous variables, while counts and percentages were calculated for categorical variables. Counts of numbers of missing values for each covariate are also reported. Counts of the uses of each antihypertensive medication for first-line, second-line, and adjunctive pharmacologic management were summarized. Prevalences, prevalence ratios, and 95% credible intervals for the prevalence ratios of each antihypertensive medication were estimated using Bayesian multinomial regression models with a logit link function.

We used informative priors for the model parameters by drawing on a multi-center study that investigated antihypertensive medication exposure in preterm infants from 1997 to 2013 across 348 NICUs managed by the Pediatrix Medical Group in the United States and clinical knowledge [11]. Priors for medication-specific parameters of the multinomial distributions were set to be Gaussian distributions with means and standard deviations that varied depending on which line of pharmacologic management was being analyzed. If a medication was not observed in our sample, we did not include that medication in the multinomial model. The complete informative priors were as follows: for first-line pharmacologic management, hydralazine's prior prevalence was set to 50%, and the prior prevalence of the other medications was split evenly between them and summed to 50%. The prior standard deviations were set to 3. For second-line pharmacologic management, the prior prevalence of propranolol was

set to 20%, the prior prevalence of metoprolol was set to 20%, and the prior prevalence of the other medications was split evenly between them and summed to 60%. The prior standard deviations were set to 1. For adjunctive pharmacologic management, the prior prevalence of enalapril was set to 10%, and the prior prevalence of the other medications was split evenly between them and summed to 90%. The prior standard deviations were also set to 1. Because the sample sizes were so small for second-line and adjunctive pharmacologic management outcomes, slightly smaller standard deviations for the priors were required to make the posterior distributions for the prevalence ratios clinically plausible.

Prevalence ratios were calculated by dividing the posterior prevalence for one medication by the posterior prevalence for the reference medication. The "brms" package in the "R" programming language was used to fit the Bayesian models [14,15].

We conducted a sensitivity analysis in the subpopulation of patients diagnosed with primary critical congenital heart disease (ICD-9 codes: 745.0, 746.7, 746.01, 746.02, 745.2, 747.41, 745.10, 745.19; ICD-10 codes: Q20.0, Q23.4, Q22.0, Q22.1, Q21.3, Q26.2, Q20.3, Q20.5) [16]. This sensitivity analysis focused on the prevalence of these antihypertensive medications as first-line pharmacologic management only due to insufficient sample sizes for subgroup analyses of second-line pharmacologic management and adjunctive pharmacologic management. Potential subgroup analyses among patients with intraventricular hemorrhage (n = 11) or necrotizing enterocolitis (n =0) also had insufficient sample sizes.

This study was determined to be Not Human Subjects Research by Auburn University and the University of Alabama at Birmingham. The data from these electronic medical records was delivered from the University of Alabama at Birmingham to Auburn University on January 4th, 2023. Study investigators never had access to information that could identify participants, either during or after data collection. The study authors are prevented from sharing the data publicly due to restrictions placed upon use of the data by the data owners. To initiate a request for the dataset, please contact ResearchData@uabmc.edu and reference request "i2b2. lds.inpatient infants receiving antihypertensives w/dx codes <37 pt set". The corresponding author (matthew_loop@auburn.edu) will do everything he can to facilitate obtainment of the data for reasonable requests.

## Results

### Description of study sample

Our study sample of possibly eligible participants from UAB consisted of 751 encounters across 146 unique patients. We excluded 168 encounters where the participant had a gestational age at birth of 37 weeks or greater, 333 encounters where the participant was not less than 365 days postnatal age at discharge or the age at discharge was missing, 12 encounters due to the presence of diagnosis code for hemangioma, 67 encounters because there was no pharmacy data available for the encounter, and 37 encounters because the participant did not receive any of the medications of interest during that encounter. These exclusions resulted in a final sample size of 134 encounters across 120 unique patients. Fig 1 shows a flow diagram of these exclusion criteria. Eighty-nine percent of patients had only one encounter, 10% had two encounters, and one patient had three encounters.

Table 1 presents the summary statistics of the study sample. Over the 134 encounters in the sample, 62% of encounters were for males, 54% of encounters were among whites, and 97% were among non-Hispanic/Latino participants. The median age at discharge was 54 days and the median birthweight was 1614 g. The percent of systolic blood pressure measurements above the 95th percentile in the 24 hours prior to the first antihypertensive medication administration was 7%. The large number of missing values for percentiles was not due to

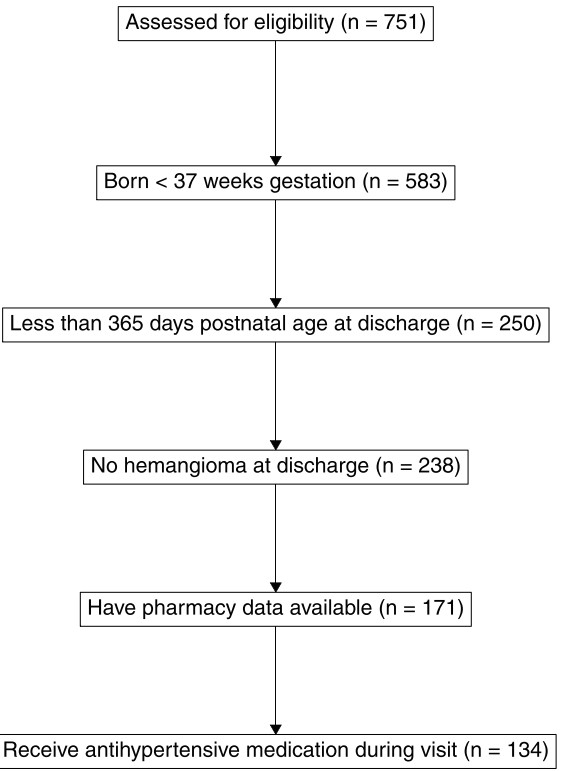

**Fig 1. Flow diagram of inclusion criteria.**

missing blood pressures, but because the percentiles we used cannot be applied to infants of certain postnatal and postmenstrual ages. Five medications (labetalol, atenolol, amlodipine, isradipine, and nifedipine) were never administered in our dataset, and follow-up with UAB confirmed that atenolol and isradipine are non-formulary at UAB.

## Summary of antihypertensive pharmacologic management

Fig 2 shows the absolute counts of antihypertensive medications used for first-line, second-line, and adjunctive pharmacologic management for the treatment of hypertension in preterm infants. First-line pharmacologic management was used in 134 encounters, second-line pharmacologic management in six encounters, and adjunctive pharmacologic management in 12 encounters. Propranolol was the most used medication for all three lines of pharmacologic management. Patients who received propranolol as first-line pharmacologic management were more likely to be male, born at an older gestational age, and have a larger birthweight than patients who received captopril as first-line pharmacologic management (S1). Fig 3 shows the posterior distributions of the prevalence of each antihypertensive medication used in each line of pharmacologic management. Among medications used as first-line medication, propranolol had the highest prevalence (61%, 95% CI: 53-69%) and hydralazine had the lowest prevalence (2.7%, 95% CI: 0.7-6%). Among medications used as second-line pharmacologic management, propranolol had the highest prevalence (40%, 95% CI: 13-71%) and sodium nitroprusside had the lowest prevalence (24%, 95% CI: 6-52%). Among medications used as adjunctive pharmacologic management, propranolol had the highest prevalence (47%, 95% CI: 23-72%) and sodium nitroprusside had the lowest prevalence (13%, 95% CI: 3-32%).

**Table 1. Demographics and baseline clinical characteristics.**

| Characteristic | N= 134[1] |
|---|---|
| Gender | |
| Female | 50 (38%) |
| Male | 83 (62%) |
| Unknown | 1 |
| Race | |
| Asian | 1 (0.8%) |
| Black or African American | 54 (43%) |
| Hispanic or Latino | 1 (0.8%) |
| Other | 2 (1.6%) |
| White | 69 (54%) |
| Unknown | 7 |
| Ethnicity | |
| Hispanic/Latino | 4 (3.3%) |
| Non-Hispanic/Latino | 117 (97%) |
| Unknown | 13 |
| Age at discharge (days) | 54 (23, 114) |
| Gestational age at birth (weeks) | 32.5 (28.4, 34.5) |
| Unknown | 27 |
| Birthweight (g) | 1,614 (890, 2,321) |
| Unknown | 70 |
| Percent of SBP*s above the 95th percentile 1 day prior to treatment (%; Dionne, Abitbol, and Flynn [2012]) | 7 (1, 21) |
| Unknown | 63 |
| Percent of SBPs above the 90th percentile 1 day prior to treatment (%; Task Force on Blood Pressure Control [1987]) | 18 (4, 37) |
| Unknown | 93 |

Data collected from inpatient visits of preterm infants treated with an antihypertensive medication at UAB from 2011 to 2022. These visits represent 120 unique patients. Patients could be included multiple times in the calculation of these summary statistics.

[1]n (%); Median (IQR); SBPs: systolic blood pressures.

Fig 4 compares the prevalence ratios of all the antihypertensive medications used across each line of pharmacologic management. Propranolol was 4.8 times (95% CI, 2.9-7.7) as prevalent as captopril, which was the second most prevalent medication used. Enalapril, hydralazine, clonidine, and sodium nitroprusside were less prevalent than captopril, whereas esmolol was similarly prevalent to captopril. Only captopril, propranolol, and sodium nitroprusside were used as second-line medications. Of these medications, propranolol was the most prevalent medication (prevalence ratio, 95% CI: 1.4, 0.3-4.3). However, the prevalences of all three medications as second-line medications were difficult to distinguish from one another given the wide credible intervals. Only esmolol, hydralazine, propranolol, and sodium nitroprusside were used as adjunctive pharmacologic management. Of these medications, propranolol was the most prevalent, and it was 3.6 (95% CI, 0.7-11.9) times as prevalent as hydralazine. However, the prevalences of all four medications as adjunctive pharmacologic management were difficult to distinguish from one another.

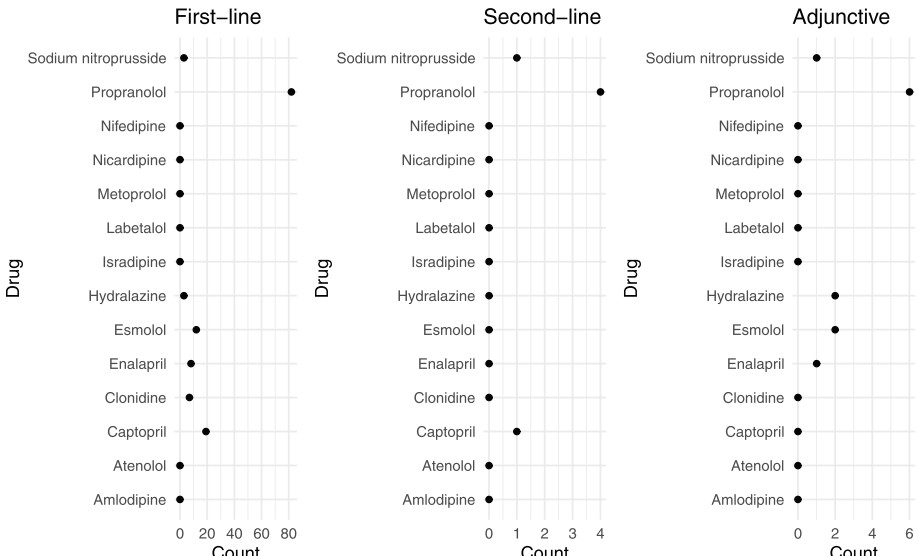

**Fig 2. Counts of medications used by line of pharmacologic management.** Propranolol was the most used medication for all lines of pharmacologic management.

## Sensitivity analysis

The sensitivity analysis focused on patients with a discharge diagnosis of primary critical congenital heart disease. S2 Table shows that patients in this subgroup were more likely to be male, were older at discharge, were born at slightly older gestational ages, and had slightly smaller birthweights than the primary analysis sample. S1 Fig shows that only propranolol, captopril, and enalapril were used as first-line pharmacologic management in this subgroup, and the prevalence of propranolol was even higher in this subgroup (80%, 95% CI: 61%-93%) than in the primary analysis. The prevalence ratio for propranolol compared to captopril was 12.4 (95% CI: 2.6-45.6).

## Discussion

The objective of this study was to determine the sequence of antihypertensive medications used for pharmacologic management of preterm infants in a single academic medical center. Propranolol was the most prevalent antihypertensive medication used as first-line, second-line, and adjunctive pharmacologic management for hypertension in preterm infants at a large academic medical center. As a beta-adrenergic blocker (beta-blocker), propranolol competitively blocks response to beta-1 and beta-2 adrenergic stimulation, which results in decreases in heart rate, blood pressure, myocardial contractility, and myocardial oxygen demand. The oral formulation of propranolol (which was used almost exclusively in our sample) is rapidly and completely absorbed with a bioavailability of approximately 25% and a half-life of 3-6 hours (Propranolol hydrochloride package insert. Major Pharmaceuticals, 2024). The most notable adverse effects associated with propranolol include bradyarrhythmias, bronchospasm, and central nervous system effects like fatigue and insomnia (Propranolol hydrochloride package insert. Major Pharmaceuticals, 2024). The results from this study suggest that the clinicians at UAB generally found propranolol's safety/efficacy profile to be favorable for pharmacologic management of hypertension in this population. Sodium nitroprusside, an intravenous (IV) vasodilator, was used as first-line, second-line, and adjunctive pharmacologic

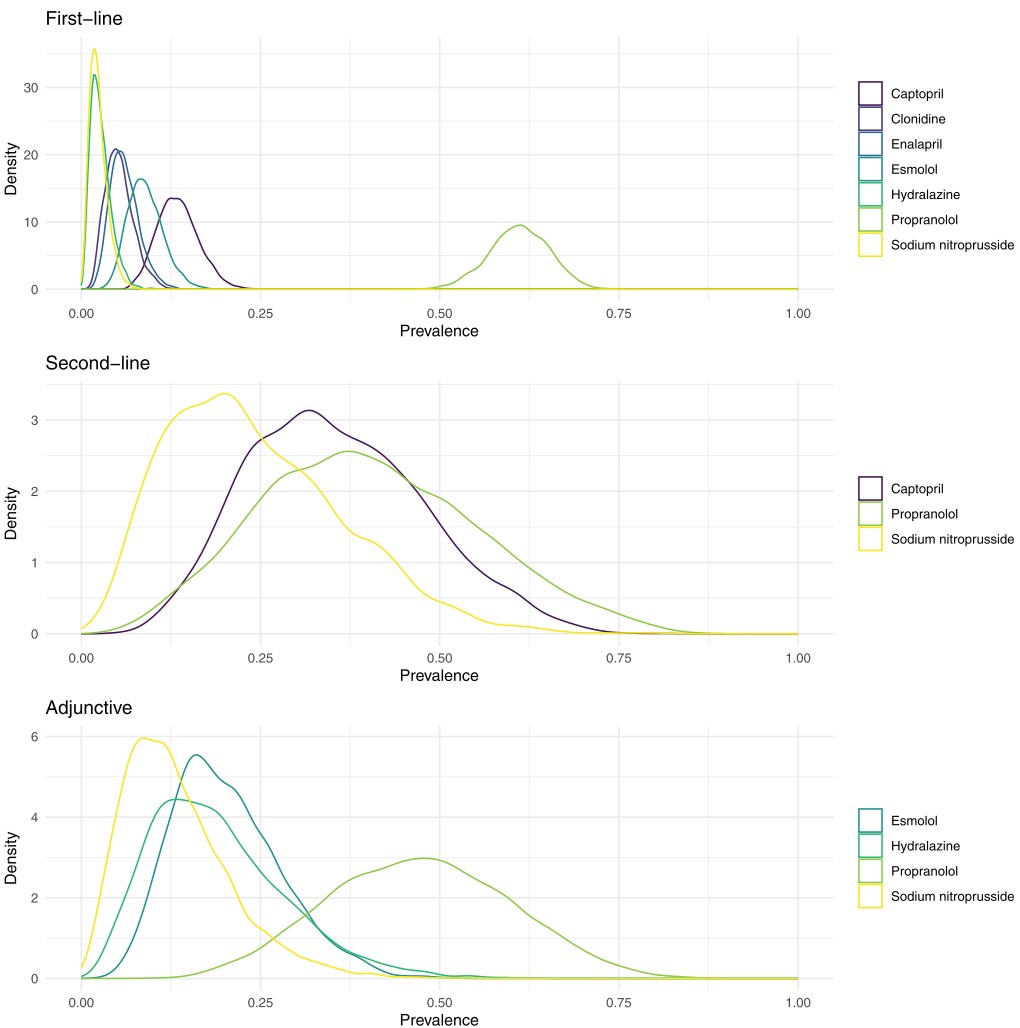

**Fig 3. Density plots comparing the posterior prevalence of each medication by line of pharmacologic management.** The inference that propranolol was the most common medication was most certain for first-line and least certain for second-line pharmacologic management.

management to varying degrees, though its prevalence relative to other medications was low in each line of pharmacologic management. Given that sodium nitroprusside has a boxed warning for both precipitous drops in blood pressure and the risk of cyanide toxicity, it is possible that clinicians at UAB may have reserved this medication for patients with more severe and/or symptomatic cases of hypertension. Clonidine, an oral alpha-2 adrenergic agonist, was used sparsely as first-line pharmacologic management and was not used at all for second-line or adjunctive medication, which suggests that clinicians at UAB may have been less comfortable with its safety/efficacy profile in this population, especially when combined with other medications. These results clarify pharmacologic management using antihypertensive medications in preterm infants with hypertension and could inform future clinical outcome-oriented studies on the topic.

The results of our study differ from the retrospective, multi-center study by Ravisankar et al. (2017), which investigated antihypertensive medication exposure in preterm infants from 1997 to 2013 [11]. In that study, hydralazine was the most commonly prescribed

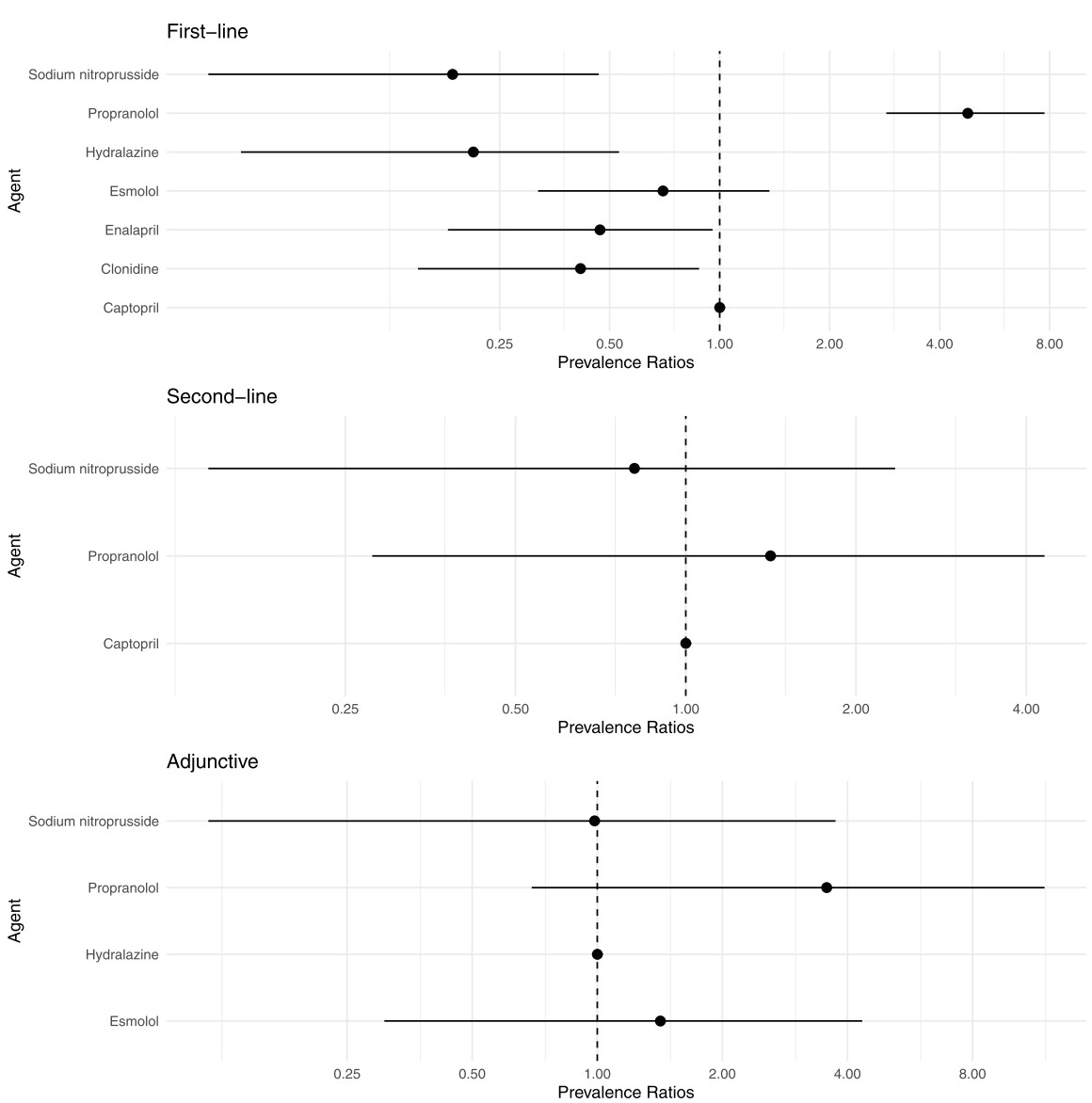

**Fig 4. Forest plots comparing prevalence ratios for each medication by line of pharmacologic management.** Propranolol was approximately 4 times more likely to be used for first-line pharmacologic management compared to captopril, the second-most used first-line medication. Large uncertainty for the prevalence ratios for second-line and adjunctive pharmacologic management made inference about the prevalence ratios inconclusive.

antihypertensive medication (51.1%), whereas beta-blockers as a class were only prescribed in 18.4% of patients. Other studies have found that calcium-channel blockers are the most commonly prescribed antihypertensive medication class [17,18]. In our study, propranolol was the most prevalent medication used for all lines of pharmacologic management. There are a variety of potential explanations for these discrepancies. Clinician preference may have shifted to beta-blockers over time, UAB may have formulary differences compared to other sites, and sample size differences may contribute to different rankings. The study by Ravisankar et al. (2017) also found that more than two antihypertensive medications were used in almost one-quarter (23.2%) of the sample [11]. In contrast, two medications were used in 13% of hospital stays in our study. There were only six administrations of a second-line medication and

only 12 administrations of adjunctive medication. A potential explanation for this discrepancy could be that the patients in this study had milder cases of hypertension compared with patients in the Ravisankar et al. (2017) study because our patients did not necessarily have a diagnosis of hypertension. Among NICU patients who receive an antihypertensive medication, 25% do not have a hypertension diagnosis [11].

Since propranolol was the most prevalent medication used in each line of pharmacologic management investigated, it would be reasonable to include propranolol (or another beta-blocker) in future studies investigating the safety/efficacy of antihypertensive medications in preterm infants with hypertension. This emphasis on propranolol may be even more justified when addressing hypertension among patients with primary critical congenital heart disease. Because propranolol was the most used medication, the cases of hypertension in this sample may have generally been mild in nature. Continuous IV infusions are generally used in emergency hypertension, where systolic blood pressure is > 5 mmHg above the 99th percentile (for patient age/weight) and the patient is experiencing systemic symptoms [6]. Another potential contributing factor to propranolol's high prevalence is that, even among patients with more severe cases of hypertension, propranolol may have been used as an adjunct to IV beta-blockers like esmolol. There were several instances in our study where esmolol and propranolol were used concomitantly, presumably to acutely lower blood pressure (with esmolol) and maintain blood pressure within a desired range (with propranolol). Another consideration for future studies could be separating oral and IV medications and investigating them separately.

One of the primary strengths of this study was its innovative dataset. We used a high-resolution dataset that provided minute-level timing of administration, dosing, and route of antihypertensive medications given. Secondly, the sample size was comparable to the existing literature on the topic [1,5], given the fact that this was a single-center study and the study population consisted of preterm infants with hypertension. Lastly, the use of Bayesian modeling and informative priors allowed us to exploit the results of other larger studies, mitigating the smaller sample size and reducing the uncertainty in our results compared to a non-Bayesian analysis.

The key limitation of this study was its single-center design. While propranolol was the medication of choice for first-line, second-line, and adjunctive pharmacologic management at UAB, this may not be the case at other health systems. There are various reasons why medication choices may vary among centers, including differences in protocols, clinician preference, and formularies. However, it is understood that there is a wide variance in how hypertensive preterm infants are managed, likely due to a lack of evidence [7,10,11]. Another limitation was that some patients may have received an antihypertensive medication for an indication other than hypertension. We accounted for hemangioma indications by excluding these patients from the study. Repeat visits from the same patient were not accounted for, but 89% of patients had only one visit. It is unlikely that this small number of repeat visits had a significant impact on our estimated credible intervals. Finally, while the overall sample size was comparable to some other studies regarding this topic, the sample sizes of patients receiving second-line and adjunctive pharmacologic management were small. Because the sample sizes were so small, our uncertainty in the posterior prevalences for each medication used for second-line and adjunctive pharmacologic management made conclusive inferences difficult.

This study's clinical takeaway is that more systematic focus on tracking how preterm infants with hypertension are treated will lead to improved outcomes for these patients. The large difference in the most common treatment for hypertension between our single-site study (propranolol) and a nationwide study of patients diagnosed with hypertension (hydralazine) [11] highlights how variable treatment practices can be among different health systems.

Different treatment practices among health systems are especially large for preterm infants [19]. Further complicating the picture is the fact that one quarter of infants who are treated for hypertension never receive a formal diagnosis, which makes the process of accurately summarizing pharmacologic management strategies difficult. Variation among health systems, and ultimately patient outcomes, can be reduced by: (i) systematically documenting the existing pharmacologic management; (ii) identifying the optimal management strategy; and (iii) producing guidelines and recommendations to disseminate evidence on optimal pharmacologic management. Since there are limited data on management strategies for preterm infants, clinicians in each health system are tasked with using their best clinical judgement. Our study extends prior research on overall exposure to antihypertensive medications and documents the sequencing of medications for pharmacological management, which is step (i) above. Future rigorous studies to build systematic evidence for the pharmacologic management of hypertension in preterm infants will lead to increased standardization of care and, ultimately, better outcomes for these patients.

## Conclusions

This retrospective, cross-sectional, single-center study identified the sequence in which antihypertensive medications were used in preterm infants at a large academic medical center. While other studies have described overall antihypertensive medication exposure in preterm infants, and review articles have outlined common regimens that are used in these patients, this study was unique in estimating the prevalences of antihypertensive medications used in each of three distinct clinical situations (first-line, second-line, and adjunctive pharmacologic management). Propranolol was determined to be the most prevalent antihypertensive medication used in each line of pharmacologic management at this single academic medical center. While this finding must be replicated to see how common this use of propranolol is across other medical centers, the results of this study could serve as a building block toward understanding the clinical decision-making surrounding the pharmacologic management of preterm infants with hypertension. This study could inform future comparative effectiveness studies investigating the safety and efficacy of antihypertensive medications used to treat hypertension in preterm infants.

## Supporting information

**S1 Table. Demographics and Baseline Clinical Characteristics of all encounters, by first-line medication received.** Represents 120 unique patients. Patients can be included multiple times in the calculation of these summary statistics.
(DOCX)

**S2 Table. Summary statistics for those included in the primary analysis (n = 134) versus those included in the sensitivity analysis of patients with a discharge diagnosis code of primary critical congenital heart disease (n = 21).**
(DOCX)

**S1 Fig. Density plots comparing the posterior prevalence of each medication in the primary analysis and in the sensitivity analysis.**
(EPS)

## Author contributions

**Conceptualization:** Alejandro D. Perez, Mary-Carty Pittman, Keia R. Sanderson, Daniel I. Feig, Matthew M. Laughon, Matthew Shane Loop.

**Data curation:** Alejandro D. Perez, Mary-Carty Pittman, Kaniz Afroz Tanni, Jieun Park, Matthew Shane Loop.

**Formal analysis:** Alejandro D. Perez, Kaniz Afroz Tanni, Jieun Park, Matthew Shane Loop.

**Investigation:** Alejandro D. Perez, Keia R. Sanderson, Matthew Shane Loop.

**Methodology:** Jieun Park, Matthew Shane Loop.

**Project administration:** Matthew Shane Loop.

**Resources:** Matthew Shane Loop.

**Software:** Jieun Park.

**Supervision:** Matthew Shane Loop.

**Visualization:** Alejandro D. Perez, Jieun Park, Matthew Shane Loop.

**Writing – original draft:** Alejandro D. Perez, Matthew Shane Loop.

**Writing – review & editing:** Alejandro D. Perez, Mary-Carty Pittman, Kaniz Afroz Tanni, Keia R. Sanderson, Jieun Park, Daniel I. Feig, Matthew M. Laughon, Matthew Shane Loop.

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
